



# Land-Use Harmonization Datasets for Annual Global Carbon Budgets

Louise Chini[1], George Hurtt[1], Ritvik Sahajpal[1], Steve Frolking[2], Kees Klein Goldewijk[3,7], Stephen Sitch[4], Raphael Ganzenmüller[6], Lei Ma[1], Lesley Ott[5], Julia Pongratz[6], Benjamin Poulter[5]

[1]Department of Geographical Sciences, University of Maryland, College Park, MD, 20742, U.S.A
[2]Earth Systems Research Center, University of New Hampshire, Durham, NH 03824, U.S.A
[3]PBL Netherlands Environmental Assessment Agency, 2500 GH The Hague, The Netherlands
[4]Department of Geography, University of Exeter, Exeter, EX4 4QE, United Kingdom
[5]NASA Goddard Space Flight Center, Greenbelt, MD, 20771, U.S.A
[6]Department of Geography, Ludwig-Maximilians University, Munich, D-80333, Germany
[7]Copernicus Institute for Sustainable Development, Utrecht University, Utrecht, The Netherlands

*Correspondence to*: Louise Chini (lchini@umd.edu)

**Abstract.** Land-use change has been the dominant source of anthropogenic carbon emissions for most of the historical period, and is currently one of the largest and most uncertain components of the global carbon cycle. Advancing the scientific understanding on this topic requires that the best data be used as input to state-of-the-art models in well-organized scientific assessments. The Land-Use Harmonization 2 dataset (LUH2), previously developed and used as input for CMIP6 simulations, has been updated annually to provide required input to land models in the annual Global Carbon Budget (GCB) assessments. Here we discuss the methodology for producing these annual LUH2-GCB updates and extensions which incorporate annual FAO wood harvest data updates for dataset years after 2015 and HYDE gridded cropland and grazing area data updates (based on annual FAO cropland and grazing area data updates) for dataset years after 2012, along with extrapolations to the current year due to a lag of one or more years in the FAO data releases. The resulting updated LUH2-GCB datasets have provided global, annual gridded land-use and land-use change data relating to agricultural expansion, deforestation, wood harvesting, shifting cultivation, regrowth and afforestation, crop rotations, and pasture management and are used by both bookkeeping models and Dynamic Global Vegetation Models (DGVMs) for the GCB. For GCB 2019, a more significant update to LUH2 was produced, LUH2-GCB2019 (https://doi.org/10.3334/ORNLDAAC/1851, Chini et al., 2020b), to take advantage of new data inputs that corrected cropland and grazing areas in the globally important region of Brazil, as far back as 1950. From 1951-2012 the LUH2-GCB2019 dataset begins to diverge from the version of LUH2 used for CMIP6, with peak differences in Brazil in the year 2000 for grazing land (difference of 100,000 km$^2$) and in the year 2009 for cropland (difference of 77,000 km$^2$), along with significant sub-national reorganization of agricultural land-use patterns within Brazil. The LUH2-GCB2019 dataset provides the base for future LUH2-GCB updates including the recent LUH2-GCB2020 dataset, and presents a starting point for operationalizing the creation of these datasets to reduce time-lags due to the multiple input dataset and model latencies.



## 1 Introduction

Human-induced changes to the Earth's carbon cycle are now known to have driven changes in the Earth's climate system, with far-reaching consequences for all components of the Earth system including the atmosphere, oceans, land, and human society. To gain improved understanding of the anthropogenic disturbances of the global carbon cycle and provide decision-support for climate policies, the Global Carbon Project has coordinated the publication of an annual carbon budget since 2005 (Le Quéré et al. 2013, Friedlingstein et al. 2019). These Global Carbon Budgets (GCBs) use models and observations to quantify

and partition annual anthropogenic emissions and associated natural sinks in the carbon cycle and provide the underlying data for use by the wider community.

For most of the historical period, land-use change activities such as deforestation for agriculture and wood harvesting were the primary sources of anthropogenic carbon emissions (Ciais et al., 2013). From 1750-2018, land use change emissions total

235+/-75 Pg C and fossil fuel emissions total 440+/-20 Pg C, with annual fossil fuel emissions surpassing land use change emissions in the mid 20th century (Friedlingstein et al. 2019). Although they are no longer the dominant source of human-generated carbon emissions, contemporary land-use emissions remain a large, and highly uncertain term in the global carbon budget and a driver of the interannual variability in the land carbon balance (Yue et al. 2020). Over the last decade (2009–2018) the Global Carbon Budget 2019 (Friedlingstein et al. 2019) estimated the net annual $CO_2$ flux from land-use, $E_{LUC}$, to

be 1.5±0.7 Gt C yr$^{-1}$ (compared with fossil fuel emissions over that same time period of 9.5±0.5 Gt C yr$^{-1}$). In addition, land-use related carbon emissions and removal are likely to become increasingly important in the future with both increasing demand for food and fiber by a growing and more affluent population and the potential adoption of climate mitigation strategies that heavily involve the terrestrial biosphere (such as biofuel crop production, afforestation and forest restoration, and more sustainable agricultural management practices, some of which specifically target the C budget) (Popp et al. ,2017).


For the annual Global Carbon Budget (GCB) publications, the net land-use carbon flux, $E_{LUC}$, (which is described in additional detail and context by Houghton 2020) is the sum of $CO_2$ fluxes from anthropogenic clearing of forests and other natural vegetation, afforestation, wood harvesting, forest degradation, shifting cultivation, regrowth of natural vegetation following wood harvest or abandonment of agriculture, and some land management activities, as well as decomposition from product

pools. The book-keeping models H&N2017 (Houghton and Nassikas, 2017) and BLUE (Hansis et al., 2015), and an ensemble of Dynamic Global Vegetation Models (DGVMs) as part of the TRENDY model synthesis project (Sitch et al., 2015), are used to compute $E_{LUC}$ and its uncertainty. $E_{LUC}$ is computed by the DGVMs as the difference between two simulations, one with land use and one without, and as a result it includes the loss of additional sink capacity from reduced forest cover, that is not included in the estimates of $E_{LUC}$ from book-keeping models. One of the ways in which BLUE differs from H&N2017 is that

it utilizes gridded historical land-use and land-use change maps from the Land-Use Harmonization 2 (LUH2) dataset (Hurtt et al., 2019a, 2019b, 2020), whereas H&N2017 makes use of national-level land-use data, primarily from FAO. Many of the

DGVMs also utilize the LUH2 dataset to prescribe the gridded historical land-use and land-use changes used by those models. The availability of LUH2 has facilitated the development of more comprehensive representations of LUC in DGVMs. Originally DGVMs represented only net LUC change at coarse gridcell resolution (typically greater than 0.5° / ~50km),

whereas LUH datasets have enabled models to account for sub-gridcell heterogeneity, i.e. shifting cultivation practices, and comprehensive LUC transitions. Arneth et al. (2017) suggests LUC emissions are likely greater than previously assumed as more LU detail is included in DGVMs.

The LUH2 dataset was developed as a required forcing dataset for the 6[th] Coupled Model Intercomparison Project (CMIP6)

Diagnostic, Evaluation and Characterization of Klima (DECK) and historical climate simulations (Meehl et al., 2014; Eyring et al., 2016), with its predecessor, Land-Use Harmonization 1 (Hurtt et al., 2006, 2011), having already been used extensively for the CMIP5 simulations (e.g., Brovkin et al., 2013, Boysen et al., 2014). A major goal and feature of the LUH2 dataset is that it harmonizes an historical land-use reconstruction with multiple future scenarios supplied by Integrated Assessment Models (IAMs). The historical reconstruction of LUH2 has been used and extended for Global Carbon Budget annual

assessments (Le Quéré et al., 2016; Le Quéré et al., 2017; Le Quéré et al., 2018; Friedlingstein et al., 2019, Friedlingstein et al., 2020) but LUH2 and its related datasets have now also been used for CMIP6 future scenarios (ScenarioMIP, O'Neill et al., 2016), the Land-Use Model Intercomparison Project (LUMIP, Lawrence et al., 2016), the Inter-Sectoral Impact Model Intercomparison Project 2b (ISIMIP2b, Frieler et al. 2017), Paleoclimate Modeling Intercomparison Project (PMIP, Jungclaus et al. 2017), the Global and North American Driver Data for Multi-Model Intercomparison (MsTMIP, Wei et al. 2014), and

the Intergovernmental Platform on Biodiversity and Ecosystem Services (IPBES, Kim et al. 2018, Rosa et al 2020). This wide usage of the LUH2 dataset provides a consistent treatment of land-use across multiple scientific studies.

LUH2 brings together multiple datasets at multiple different spatial and temporal scales and harmonizes them to produce a single dataset of global, gridded land-use and all associated land-use transitions, annually for the years 850-2100, in a consistent

format for use in Earth System Models (ESMs). All data layers are provided as fractional values within 0.25° spatial grids and include the area of 12 different land-use states, the age and biomass density of secondary land, wood harvest area and biomass, and agricultural management practices such as irrigated area and synthetic nitrogen fertilizer application. For the historical period the LUH2 dataset uses inputs from HYDE 3.2 (Klein Goldewijk et al., 2017) for gridded cropland, grazing land (subdivided into managed pasture and rangelands), and urban area, as well as population data, and uses national wood harvest

production data from FAO (2020a).

Early versions of LUH datasets have been used in annual GCBs from 2013 onwards (Le Quéré et al. 2014, 2015a, 2015b, 2016, 2018a, 2018b, Friedlingstein et al. 2019). For use in CMIP6 simulations, the LUH2 historical-period dataset (LUH2 v2h) was developed for the years 850-2015 and this LUH2 v2h dataset (where 'h' is historical) was extended for use as an

input into the Global Carbon Budget 2017 by incorporating updated data from FAO and HYDE for the years 2013-2014, as



well as a linear extrapolation to the year 2016, as described in Methods (note: the GCB includes a projection for the current year, but is based on model simulations and thus forcing until the previous year, i.e. GCB 2017 incorporates LUH2 data until 2016). The intended strategy for subsequent annual LUH2-GCB updates was to use the significant updates to LUH that are typically made on the same cycle as CMIP analyses, and then provide annual extensions to those each year, while also ingesting

any new/modified data from FAO and HYDE for recent years. Although this strategy involves multiple dataset and model update latencies, it was successfully used to extend the LUH2 v2h dataset for GCB 2017 and GCB 2018 with updated FAO and HYDE inputs, along with extrapolations for years without underlying data. However, the LUH2-GCB2019 dataset (Chini et al., 2020b) provided a more significant update that took advantage of new input datasets and corrections of previous inputs for Brazil, which is a region of significance for the global carbon budget. This was the first major update to the LUH-GCB

datasets since the development of the LUH2 v2h dataset, and it resulted in small changes from 1951 onwards and more significant modifications to the cropland area trajectories during the 2000-2010 period and grazing land trajectories during the 1990-2010 period.

With multiple versions of LUH2 datasets now being used for different synthesis studies, and multiple versions of key inputs

to LUH2 also in existence, it is important to have a documented record of these datasets and their intended uses, along with the methodology underlying each of them. In this paper we will describe and document the general approach for generating the LUH-GCB datasets, as well as specific details of the recent changes made to the LUH2-GCB2019 dataset, and comparisons with previous LUH datasets.

## 2 Methods

### 2.1 Land-Use Harmonization 2 Dataset

The Land-Use Harmonization 2 dataset (Hurtt et al. 2019a, 2019b) provides global, annual, gridded land-use states and all associated land-use transitions between those states, fractionally, at 0.25° spatial resolution for the years 850-2100 (Hurtt et al. 2020). The twelve LUH2 land-use states include gridded fractions of cropland (which is further sub-divided into fractions of C3 annuals, C4 annuals, C3 perennials, C4 perennials, and C3 N-fixers), grazing land (sub-divided into fractions of managed

pasture and rangeland), urban land, primary land (both forested and non-forested) and secondary land (both potentially forested and potentially non-forested). In addition, the LUH2 dataset provides estimates of the age and biomass density of secondary lands (estimated from the empirical Miami model of net primary production, Leith 1972), gridded areas and biomass associated with wood harvest, and gridded cropland management data including irrigated areas and synthetic nitrogen fertilizer application rates (all provided at the same spatial and temporal scales as the land-use states and transitions). As described in

Hurtt et al. 2020, the LUH2 dataset was computed with the Global Land-Use Model 2 (GLM2) which harmonizes multiple input datasets at multiple different temporal and spatial scales to produce a single land-use dataset in the format required for use in climate, carbon, and vegetation models. LUH2 includes major updates to several features, including a new historical

national wood harvest reconstruction (Kaplan et al. 2017), a new representation of shifting cultivation (Heinimann et al. 2017), use of Landsat remote sensing data (Hansen et al., 2013) to constrain patterns of wood harvesting, crop rotations, and connects

to 8 different future scenarios provided by Integrated Assessment Models (O'Neill et al., 2016, Popp et al., 2017). The version of GLM2 that was used for generating the LUH2-GCB2019 dataset is identical to that used for generating the LUH2 dataset, with the exception of some minor corrections to the method for determining the spatial patterns of wood harvesting.

For the historical period (years 850-2015), the LUH2 dataset was based on two key inputs: the HYDE 3.2 dataset and FAO
wood harvest data. The HYDE 3.2 dataset (Klein Goldewijk et al. 2017) provides gridded cropland, managed pastureland, rangeland, and urban land data, and is itself based on national FAO agricultural data for the years 1961-2012, along with extrapolations based on a 5-year trend for the years 2013-2015. The FAO wood harvest data provides national forestry statistics for the years 1961-2015 (FAO 2020a) that were used by LUH2 to determine the spatial patterns of wood harvesting.

## 2.2 General approach for annual LUH extensions for GCB

Each year, for the annual Global Carbon Budget (for publication years 2013-2020), the LUH dataset was extended in time for use in the participating DGVM and book-keeping model simulations. In general, these LUH extensions were built off the most recent LUH release (i.e. either LUH1 or LUH2) and incorporated updated and extended agricultural area data from HYDE (based on newly released data from FAO) along with newly released national wood harvest production data from FAO. This not only extended the LUH datasets beyond their final year (2005 for LUH1 and 2015 for LUH2) but also updated the LUH
data for years that were not previously based on FAO inputs (i.e. years for which data was previously extrapolated but for which FAO data was now available).

The annual updates for GCB from the HYDE dataset were updates to the most recent HYDE release, either HYDE 3.1 (Klein Goldewijk et al., 2010, 2011) or HYDE 3.2 (Klein Goldewijk et al., 2017). Both HYDE 3.1 and HYDE 3.2 used data from
FAO to inform the spatial patterns and areas of cropland, grazing land, and urban land for years in which FAO data was available, and used extrapolations and additional datasets to inform the years after the last "FAO-data-based" year. When producing annual updates to HYDE using the most recent FAO data releases, the new FAO data could often not be simply appended to the previously used FAO data due to the retrospective changes to the FAO datasets that sometimes occur. To resolve this challenge, the annual updates to HYDE 3.1 or HYDE 3.2 used an anomaly approach to update the cropland,
grazing land, and urban land grids by applying the annual differences in new FAO data to the last "FAO-data-based" year of the current HYDE dataset. The updated HYDE data was also extrapolated using a 5-year trend for years without underlying FAO data. HYDE 3.2 was provided to LUH2 each decade from 1700 to 2000 and then annually for the years 2000 onwards.



The national wood harvest reconstruction used by LUH was updated with new data from FAO by replacing any extrapolated wood harvest data from the current LUH dataset with the new wood harvest data from FAO, and then extrapolating national wood harvest rates for any remaining years without underlying FAO data.

The schematic diagram in Figure 1a shows the interactions and dependencies between the LUH datasets prepared for use in CMIP, the LUH-GCB datasets prepared for use in GCB simulations, as well as the inputs used by LUH that are typically updated on both CMIP as well as annual cycles. The FAO data that informs the HYDE dataset and the LUH national wood harvest reconstruction typically lag behind the current year by one or more years. As a result, the LUH-GCB datasets contain several data latencies and data extensions. A typical timeline for this process is shown in Figure 1b, illustrating the sequence of dataset updates, and associated latencies, that occur during the creation of a generalized 'LUHx-GCBi' dataset.

For GCB publication years 2013-2015, the LUH-GCB dataset was built off the LUH1 dataset and was identical to that dataset for years 1500-2005. The data for the years 2006 to the end of the LUH-GCB timeseries was based on new data from FAO and HYDE, using the aforementioned anomaly approach (for years that FAO data existed at the time of dataset creation), and extrapolations for years without FAO data. Table 1 provides the specific years that are based on updated FAO and HYDE data in each of these LUH-GCB datasets. For LUH data years without specified FAO data, extrapolations were used to fill data gaps.

For GCB 2016, an updated LUH dataset was not provided due to ongoing work on the creation of the LUH2 dataset. Instead, GCB used only those DGVMs that were based directly on HYDE, not LUH (Le Quéré et al., 2016).

For GCB publication years 2017-2018, the LUH-GCB dataset was built off the LUH2 dataset and was identical to that dataset for years 1500-2012 (i.e. to the end of the time period for which FAO data was used in the creation of HYDE 3.2). The data for the years 2013 to the end of the LUH-GCB timeseries was based on new data from FAO wood harvest and HYDE, again using the aforementioned anomaly approach, along with extrapolations for years without underlying FAO data, as outlined in Table 1.

For GCB publication years 2019-2020, a more significant update was performed to correct an error in the previously used input datasets, especially for the country of Brazil, as described in Section 2.3. These LUH-GCB datasets were built off the LUH2 dataset and were identical to that dataset for all years up to and including the year 1950. The data for the years 1951 to the end of the LUH-GCB timeseries was based on new data from FAO (FAO 2020b) and corrections from HYDE, along with extrapolations for years without underlying FAO data, as outlined in Table 1.





### 2.3 HYDE land-use data update for GCB 2019

The HYDE-GCB2019 dataset used for LUH2-GCB2019 corrected some inadvertent errors present in its underlying input datasets, specifically the sudden leveling-off of cropland area expansion around the year 2000 and a compensating sudden increase in cropland-related clearing of natural vegetation in Brazil around the year 2010. Similarly, an overestimate of grazing

land area in Brazil between 1990 and 2010 was eliminated. HYDE-GCB2019 was based upon the HYDE3.2 (as were HYDE-GCB2017-18), although using a slightly different variant of HYDE 3.2 from that used in LUH2 v2h. In addition, HYDE-GCB2019 incorporated updated cropland and grazing land data from FAO as far back as 1961, through to 2015, as well as sub-national data for Brazil from Instituto Brasileiro de Geografia e Estatística (IGBE) for the years 1920-1995, and European Space Agency (ESA)-derived data for the year 2010. In addition, the HYDE-GCB2019 dataset was extrapolated to 2019 using

a 5-year trend in gridded land-use areas (2014-2018).

Despite the differences between the HYDE3.2-Aug2016 used for LUH2 v2h and the HYDE-GCB2019, for each year up to and including 2012, the HYDE-GCB2019 was closely matched to the HYDE3.2-Aug2016 dataset, with the exception of grid-cells in the corrected regions of Brazil. The global areas provided by the HYDE-GCB2019 dataset differ from the HYDE3.2-

Aug2016 dataset by up to 0.5% for cropland and up to 0.3% for grazing land. However, within Brazil the areas in HYDE-GCB2019 differ from those in HYDE3.2-Aug2016 by up to 11% for total cropland (in 2009) and up to 5% for total grazing land (in the year 2000). In the year 1960 (the first year of corrected data in HYDE-GCB2019 and the year in which we would expect the closest match between the two datasets) 0.4% of grid-cells (at 5 minute spatial resolution) outside Brazil had cropland area differences of greater than 10% from the HYDE3.2-Aug2016 data, and 0.01% had grazing land area differences

of greater than 10% from the HYDE3.2-Aug2016 data.

### 2.4 National wood harvest data update for GCB 2019

The version of wood harvest data used for LUH2 v2h was based on a previous FAO release (FAO 2020a) that included data up to and including the year 2014 – those inputs remained the same in this new GCB dataset. For LUH2-GCB2019, the most recent FAO data (FAO 2020b) was used to provide wood harvest data for the years 2015-2017. The annual changes in FAO

wood harvest data for the years 2015-2017 were applied to the year 2014 data from the previous release to get the new 2015-2017 data used for GCB 2018. After the year 2017 wood harvest data was extrapolated to the year 2018, the last year in which GLM2 simulates wood harvest for LUH2-GCB2019.

### 2.5 Computing the LUH2-GCB2019 dataset

To enable long-term historical simulations, the LUH datasets must provide a consistent and continuous data time-series over

the entire simulation time domain (i.e. from 850 onwards). This implies that annual dataset updates must be harmonized with the core version of LUH being used (either LUH1 or LUH2) at the point in time in which the dataset extension begins to

diverge from the core LUH version. HYDE provides decadal data for years from 1700 up to 2000, and the data in HYDE-GCB2019 was different to that in HYDE3.2-August2016 (the version used for LUH2 v2h) from 1960 onwards, making 1950 the most recent year of identical data between these two datasets and the "harmonization" year between the LUH2 core dataset

and the LUH2-GCB2019 update.

The HYDE-GCB2019 dataset for the years 1960-2019, along with the HYDE3.2-August2016 data for the year 1950, were aggregated to 0.25° spatial resolution and interpolated to annual time-steps, to create the cropland, grazing land, and urban land input datasets for LUH2 for the period 1950-2019. The GLM2 code (Chini et al., 2020a), used to generated the LUH2

datasets, was then run with the processed HYDE data, and the FAO-based national wood harvest data, as inputs for the years 1950 to 2019, using the LUH2-v2h data for the year 1950 as an initial condition. Using this approach, the LUH2-GCB2019 data for the years 1950-2019 connected continuously to the LUH2-v2h dataset for the years 850-1950.

## 3 Results

The LUH2-GCB2019 dataset shares many of the same properties as its predecessors LUH2 v2h, LUH2-GCB2017, and LUH2-

GCB2018, and is identical to those datasets for all years up to 1950. Figure 2a shows the global area of the five aggregated land-use states represented by LUH2-GCB2019 for the years 1950-2019. Global cropland area increased from $12.2 \times 10^6$ km$^2$ in 1950 to $15.9 \times 10^6$ km$^2$ in 2015 (for LUH2 v2h) or $16.1 \times 10^6$ km$^2$ in 2015 (for LUH2-GCB2019) and $16.8 \times 10^6$ km$^2$ in 2019 (for LUH2-GCB2019). Global grazing land area increased from $26.1 \times 10^6$ km$^2$ in 1950, peaked in the year 2000 at $33.2 \times 10^6$ km$^2$ (for LUH2 v2h) or peaked in 2001 at $33.1 \times 10^6$ km$^2$ (for LUH2-GCB2019), then decreased to $32.8 \times 10^6$ km$^2$ in 2015 (for

LUH2 v2h) or $32.7 \times 10^6$ km$^2$ in 2015 (for LUH2-GCB2019) and $32.6 \times 10^6$ km$^2$ in 2019 (for LUH2-GCB2019). Cropland, grazing land, and urban land areas in LUH2-GCB2019 were identical to those areas provided by the HYDE dataset at global, regional, and 0.25 degree spatial scales (by design).

Looking at Brazil specifically, Figure 2b shows that the new input data underlying LUH2-GCB2019 has modified the

previously observed feature of LUH2 v2h in which cropland area anomalously leveled-off, and then quickly increased, during the 2000-2010 time-period. The cropland area over the same time period in LUH2-GCB2019 now increases monotonically. Similarly, LUH2-GCB2019 has also reduced the area of grazing land in Brazil between the years 1990 and 2010, when compared with the LUH2 v2h grazing land area.

Land-use states began to differ between LUH2-GCB2019 and LUH2 v2h from 1951 onwards. Figure 2c shows the differences in global areas of the five LUH2 land-use states between these two datasets from 1950-2015 (solid lines). Although differences in cropland and grazing land areas didn't become significant until 1990 onwards, differences in primary and secondary land areas appeared in the time series from around 1960 onwards, due to slightly different methodology for determining the spatial



patterns of primary vs. secondary wood harvesting in some locations of the world, compared with the methodology used in preparation of the LUH2 dataset. Figure 2c also shows the same variables but focused only on the country of Brazil (dashed lines). Prior to 2010, the differences between the LUH2 v2h and LUH2-GCB2019 datasets in Brazil were of similar magnitude to the global differences in the five land-use variables, indicating that the changes made to the LUH2-GCB2019 dataset within Brazil account for the majority of the global differences in this dataset. Cropland and grazing land differences between the LUH2 v2h and LUH2-GCB2019 datasets became largest during the 2000-2019 timeframe when input datasets (based on FAO) began to diverge more significantly, especially in the country of Brazil. The difference in global cropland area between the two datasets has a low point in 2009 of -77,500 km$^2$ of which 99% comes from cropland differences within Brazil. The maximum difference in global cropland area of -203,900 km$^2$ occurred in 2015. The difference in global grazing land area between the two datasets was highest in year 2000 with a difference of +102,600 km$^2$, of which 98% comes from grazing land differences within Brazil.

Figure 3 shows that global wood harvest, one of the dominant land-use transitions in the LUH2 dataset, remained largely unchanged between LUH2 v2h and LUH2-GCB2019, with global wood harvest differences being less than $2\times10^{-3}$ PgC (or <0.2%) each year. Total gross transitions (the sum of the absolute value of all land-use transitions) are a measure of all land-use change activity. Gross transitions in the LUH2 dataset tend to have higher interannual variability after year 2000 when the input data to LUH2 (from HYDE) becomes annual, rather than decadal. Figure 4a shows that global gross transitions between 1950 and 2012 (years that are based on the same FAO data) were very similar between the LUH2 v2h and LUH2-GCB2019 datasets, with maximum global annual differences of less than 3%. After 2012, differences between the two datasets increased slightly due to the use of different FAO data in the HYDE dataset, but global differences still remained less than 10%. Gross transitions within Brazil (Figure 4b) showed more significant differences between these two datasets, with the largest absolute differences occurring in 2008 (-52,000 km$^2$/yr difference) and 2009 (47,000 km$^2$/yr difference), which was consistent with the timeframe of the maximum cropland area differences within Brazil.

The spatial patterns of differences (of the fractional content of each quarter degree grid-cell) between the LUH2 v2h and LUH2-GCB2019 datasets for 3 key variables at 3 different time points are shown in Figure 5. The cropland patterns illustrate that in the year 2000 and 2009 almost all grid-cell differences were located within Brazil, with grid-cells elsewhere having only very minor differences, if any (fraction of grid-cell of $\sim10^{-3}$). Although the total difference in Brazil cropland area between the two datasets in 2000 was very small (504 km$^2$), Figure 5 shows that sub-national differences existed that both increased cropland in some sub-regions of Brazil, and reduced it in others (and aggregated to a small national total difference). In the year 2009, when the Brazil total cropland difference between LUH2 v2h and LUH2-GCB2019 was greatest, the gridded differences in Brazil were almost all in the same direction, which in aggregate led to a 77,000 km$^2$ correction from the LUH2 v2h values. In 2015 gridded cropland differences in Brazil were small (and the aggregate national difference for Brazil was also small) but significant gridded differences in other locations in the world appeared, especially within China, Southeast



Asia, and Australia. These differences were due to new FAO data for the years 2013-2015 being used by the HYDE dataset within LUH2-GCB2019 (compared with the extrapolated data used for LUH2 v2h) and represent the error in previous
projections/extrapolations before data was available.

The patterns for grazing land in Figure 5 show that in the year 2000, although the country of Brazil had a total grazing land decrease of 10,000 km$^2$ in LUH2-GCB2019 (compared with LUH2 v2h), there were sub-regions within Brazil that were exhibiting increases in grazing land that offset some of these decreases. In the year 2009, grazing land differences between
LUH2 v2h and LUH2-GCB2019 were small (fractions of grid-cell ~10$^{-3}$), both within Brazil and elsewhere in the world. In 2015, grazing land differences between the datasets showed larger differences throughout the world, due to the use of new FAO data for the years 2013-2015, as was also observed with cropland, and again represent errors in previous projections/extrapolations.

Secondary forest patterns in Figure 5 showed that LUH2-GCB2019 had greater secondary forest area in high latitude locations, as well as in China and Southeast Asia, and lower areas of secondary forest in the Amazon and Congo regions, across all time periods after 1950. These changes in secondary forest patterns reflected the related changes to the updated GLM2 methodology for the spatial patterns of wood harvesting, which resulted in more primary wood harvest in the high latitudes, generating more secondary land, with the converse occurring in the Amazon and Congo regions.

**4 Discussion**

The LUH2-GCB2019 dataset (Chini et al., 2020b) is the annual update to the LUH2 dataset for use as input to the ensemble of international DGVMs (TRENDY DGVM model synthesis, Sitch et al., 2015) that delivers to (and supports) the 2019 Global Carbon Budget. It was based on existing LUH2 methodology (Hurtt et al. 2020) with updated inputs from HYDE (for gridded cropland, grazing land, and urban land areas at decadal and/or annual timesteps) and from FAO (for annual national wood
harvest amounts). In contrast with previous LUH2 annual updates for GCB that incorporated new data from HYDE and FAO for the years 2013 onwards and extrapolated the time-series to the current year, the LUH2-GCB2019 dataset provided a more significant update to correct the anomalous decrease in cropland area and overestimate of grazing land within Brazil over the years 2000-2009 due to errors in the previously used FAO and HYDE data for Brazil. This correction was successful and resulted in a national cropland area difference for Brazil of up to 77,000 km$^2$ in the year 2009, and also modified the national
grazing land area within Brazil over the years 1990-2009 with a difference of up to 100,000 km$^2$ in the year 2000. Differences between LUH2-GCB2019 and LUH2 v2h in other regions of the world were more minor, as shown in Figures 2 and 5, with the exception of small differences in the spatial patterns of wood harvesting that resulted in an increase of secondary forest area in the northern latitudes, and a reduction of secondary forest area in the Amazon and Congo regions.



Global gross transitions (a measure of all land-use change activity) were very similar between LUH2 v2h and LUH2-GCB2019, although slightly higher for LUH2-GCB2019 after the year 2000. In contrast, the gross transitions for Brazil tended to be lower for LUH2-GCB2019 compared with those for LUH2 v2h, with the exception of the 2000-2009 period when gross transitions within Brazil for LUH2-GCB2019 became much larger than those in LUH2 v2h. Those land-use transitions in the early-2000s, when the main correction to the Brazil land-use data was applied, are likely to provide the biggest changes to the

global carbon budget. The high annual variability in land use transitions after year 2000 was an artifact of the input data from HYDE, which was provided to LUH2 each decade from 1700 to 2000 and then annually for the years 2000-2019, and does not necessarily indicate that land-use transitions were fluctuating more quickly in the years 2000-2009 when compared with the earlier part of the historical period.

The differences between the LUH2-GCB2019 dataset and previous versions have implications for the global and regional carbon budgets, and many models that were part of GCB2019 have now used the LUH2-GCB2019 dataset to compute the carbon fluxes associated with land-use activities. For example, the bookkeeping model BLUE (Hansis et al., 2015) has computed the land-use emissions from both the LUH2-GCB2019 and LUH2-GCB2018 datasets, and compared them globally and regionally. The results from BLUE indicate that global carbon emissions from land-use from over the years 2000 to 2009

are similar, although slightly higher, in LUH2-GCB2019 than in LUH2-GCB2018, with a peak difference in 2008 of 340 Tg C yr$^{-1}$, and with the majority of these differences attributed to emissions from expansion of cropland. LUH2-GCB2019 gives higher land-use emissions from BLUE in Brazil over that time period, with large differences for emissions from cropland expansion, but only small differences from expansion of grazing land, from wood harvesting, or from vegetation regrowth. The higher emissions from 2000-2009 in LUH2-GCB2019 were expected, due to the additional cropland expansion in LUH2-

GCB2019. However, global emissions from BLUE over the period 1960-1980, as well as those in Brazil over the same time period, are slightly lower in LUH2-GCB2019 when compared with LUH2-GCB2018, despite very small net differences in cropland area between LUH2-GCB2019 and LUH2-GCB2018 over the time period, both globally and within Brazil. Gross transitions within Brazil from 1960-1980 are also lower in LUH2-GCB2019 when compared with LUH2-GCB2018. Inspection of the spatial patterns of cropland area differences between 1960 and 1980 for both LUH2-GCB2019 and LUH2-GCB2018

shows significant sub-national reorganization of cropland areas between the two different datasets within Brazil, resulting in a very small net difference in cropland area, but potentially driving lower gross transitions and lower cropland-related emissions in LUH2-GCB2019 over that time period.

In addition to its usage in the 2019 Global Carbon budget and associated TRENDY DGVM model synthesis, the LUH2-

GCB2019 has also been used as part of a seasonal carbon forecasting project (Ott et al., 2018) in which the LUH-GCB methodology is used, in the absence of existing forecasting datasets, to give land-use projections. As part of this project, the LUH2-GCB2019 land-use data was compared with previous projections and estimates as well as the Shared Socioeconomic Pathways (Popp et al. 2017). Due to the inclusion of new FAO data, the global LUH2-GCB2019 cropland and grazing land





areas have a constant offset from previous land-use projections, which provides a measure of the skill of those previous
projections. The global LUH2-GCB2019 cropland and grazing land trajectories are comparable with many of the SSP future
scenarios, although they contain significant regional and sub-national differences, which will be explored in more detail.

The LUH2 land-use changes/transitions are typically converted into land-cover changes by DGVMs for use in those models.
DGVM groups have often asked for guidance on which land-cover change(s) to associate with each LUH2 land-use change,
and in particular have asked for a recommendation on whether natural vegetation should be lost when converting to managed
pasture and/or rangeland. Based on recommendations from HYDE, as well as the study by Ma et al. 2020, the LUH2
recommendation is that all natural vegetation should be cleared for managed pasture, and only cleared for rangeland if it is
forested (Hurtt et al. 2020). Using this rule/guideline gives maps of forest area, carbon density, and carbon emissions that are
consistent with other published maps.


A challenge when producing the annual LUH2 updates for the Global Carbon Budget is handling the inherent latencies
associated with input data updates and model developments that exist in this process, as illustrated in Figure 1. FAO wood
harvest data is updated annually, but typically lags behind the current year by 2 years, requiring extrapolation to complete the
time-series. FAO cropland and grazing land data (used by HYDE) is updated less frequently and sometimes lags behind the
current year by several years. Previously published values for existing years in the FAO datasets can also be modified
retrospectively as part of the FAO dataset updates, resulting in dataset inconsistencies. In addition to data input updates, the
models themselves are also constantly evolving. The model underlying the HYDE dataset is updated frequently to incorporate
new data, to use new methodology for computing gridded land-use states, and to provide additional data layers, while the
LUH2 model itself is typically updated for every CMIP cycle with minor updates and bug fixes in between those updates.
These widely varying timescales for updates to data and models present some challenges for providing a consistent, yet
annually updated dataset for large model synthesis projects such as the Global Carbon Budget, while also making use of the
latest data and model improvements, and our ongoing goal is to continue reducing these latencies wherever possible. Our
approach thus far has involved a desire to provide consistency with climate simulations and with previous GCBs and to enable
comparison across scientific studies that are utilizing the same underlying land-use data. As a result we have avoided
incorporating major updates to the LUH methodology and input datasets between CMIP updates. However, the downside of
this approach is that important input-data updates do not always appear in the LUH-GCB datasets in a timely manner. This
tension between consistency and incorporating new data requires a balanced approach, which we have attempted to achieve in
the LUH2-GCB2019 update in which we incorporated more significant dataset corrections beyond the regular extensions. An
aspirational goal for these LUH-GCB products, would be to update the datasets in a more operational mode, in which model
and input-data updates would be automated annually. This would require a centralized activity, including operational support
and coordination, between the FAO, HYDE, and LUH teams. Although this goal would undoubtedly present its own challenges

and competing constraints, it would enable an important input dataset for the Global Carbon Budget to be updated consistently and collaboratively over the long term.

Subsequent to generating the LUH2-GCB2019 dataset, we also recently generated the LUH2-GCB2020 dataset for use in GCB2020 simulations (Friedlingstein et al. 2020). This dataset used LUH2-GCB2019 cropland, grazing land, and urban land grid-cell fractions for years up to and including 2019, and extrapolated those land-use states to 2020 using a 5-year trend from HYDE. The national wood harvest inputs used for LUH2-GCB2019 were used for all years up to and including 2014, after which new wood harvest data from FAO was used, including extrapolations for the years 2019 and 2020. The differences in
wood harvest inputs for the years 2015-2019 between LUH2-GCB2019 and LUH2-GCB2020 resulted in some small differences for primary and secondary land fractions for those years.

Looking ahead, we anticipate providing an LUH2 update for use in the Global Carbon Budget 2021 simulations which will likely incorporate new HYDE and FAO data for very recent years and extend the time-series to 2021. In addition, with plans
already beginning for CMIP7, we can anticipate the need for LUH3. While it is too early to define the priorities for this dataset, we can anticipate some of the potential features. The development could include a new version of the HYDE dataset, new future scenarios, the use of additional satellite remote sensing data, as well as the introduction of additional land-use/land management  processes such as forest degradation and plantation forestry.

**Code Availability**

The source code used to produce the core LUH2 datasets as well as the LUH2-GCB datasets, along with the sources and citations of necessary inputs, are archived at http://doi.org/10.5281/zenodo.3954113 (Chini et al., 2020a).

**Data Availability**

The data produced in this study are archived and publicly available at the NASA Oak Ridge National Laboratory Distributed Active Archive Center: https://doi.org/10.3334/ORNLDAAC/1851 (Chini et al, 2020b).

**Author Contributions**

LC is the lead author and co-developed the method and conducted analyses with GH, RS, SF, SS, and KKG. KKG provided historical input data. RG provided data from the BLUE book-keeping model. LM, LO, JP, and BP provided modeling input. All authors contributed to writing the manuscript.



**Competing Interests**

The authors declare that they have no conflict of interest.

**Acknowledgements**

We gratefully acknowledge the support of NASA grants 80NSSC17K0348 (NASA-IDS) and 80NSSC17K0710 (NASA-CMS). This research was also supported as part of the Energy Exascale Earth System Model (E3SM) project, funded by the U.S. Department of Energy, Office of Science, Office of Biological and Environmental Research.

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



| Publication / Dataset | LUH base version | LUH-GCB simulation years | HYDE base version | FAO agricultural data | FAO wood harvest data |
|---|---|---|---|---|---|
| LUH1 | LUH1 | 1500-2005 | 3.1 | 1961-2005 | 1961-2005 |
| GCB 2013 | LUH1 | 1750-2012 | 3.1 | 1961-2010 | 1961-2011 |
| GCB 2014 | LUH1 | 1750-2013 | 3.1 | 1961-2010 | 1961-2012 |
| GCB 2015 | LUH1 | 1750-2014 | 3.1 | 1961-2012 | 1961-2013 |
| GCB 2016* | None | | | | |
| LUH2 | LUH2 | 850-2015 | 3.2*** | 1961-2012 | 1961-2015 |
| GCB 2017** | LUH2 | 1750-2016 | 3.2 | 1961-2014 | 1961-2015 |
| GCB 2018 | LUH2 | 1750-2018 | 3.2 | 1961-2015 | 1961-2016 |
| GCB 2019 | LUH2 | 1750-2019 | 3.2 | 1961-2015 | 1961-2017 |
| GCB 2020 | LUH2 | | 3.2 | 1961-2015 | 1961-2018 |



**Table 1. Input datasets and simulation years for annual LUH-GCB datasets and core LUH products**
**\* LUH data not used while new LUH2 dataset was under development**
**\*\* First year LUH used for BLUE as well as for DGVMs/uncertainty**
**\*\*\* The specific version of HYDE 3.2 that was used for LUH2 is the August 2016 beta release**




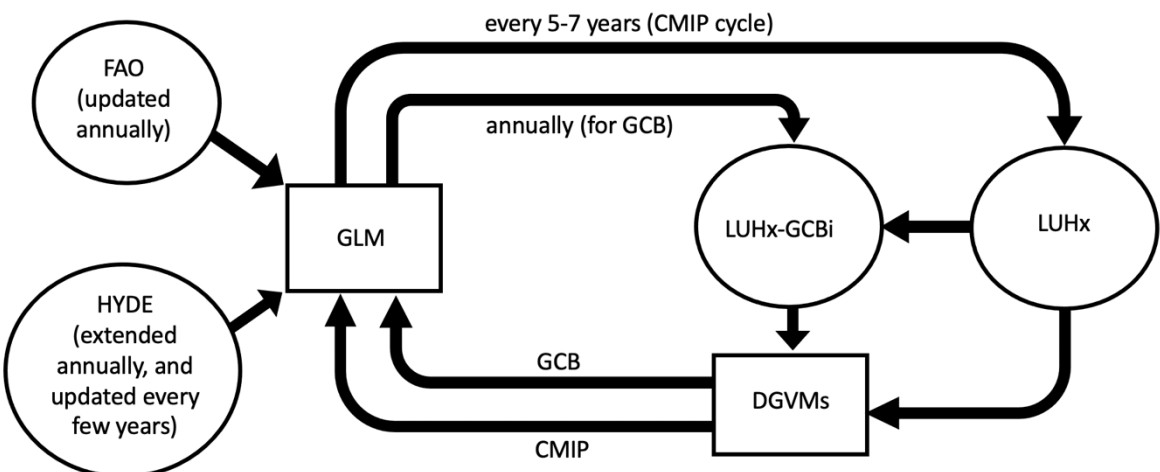

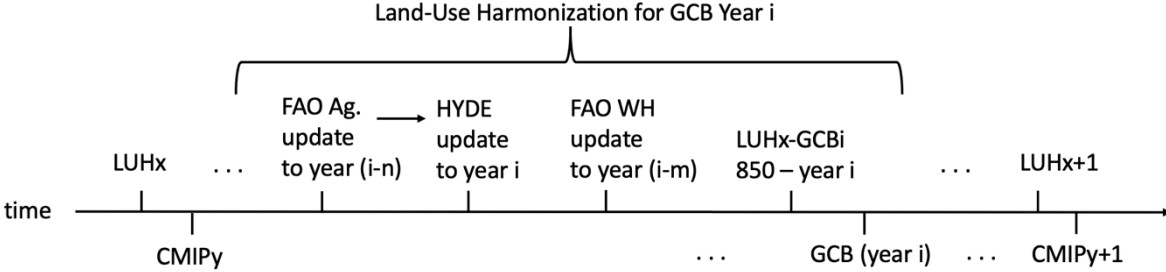

**Figure 1: a) Schematic diagram of LUH-GCB annual updates showing interactions and dependencies between core LUH products, LUH-GCB datasets, and annually-updated inputs, b) timeline of LUH-GCB annual updates illustrating the sequence of dataset updates and associated latencies that occur during the creation of a generalized LUHx-GCBi dataset.**



**Figure 2. a) Global area of key land-use states in LUH2-GCB2019, b) Area of cropland and grazing land in Brazil in both LUH2-GCB2019 (solid line) and LUH2 v2h (dashed line), c) Differences between global areas (solid lines) and Brazil areas (dashed lines) of key land-use states (LUH2 v2h − LUH2-GCB2019).**




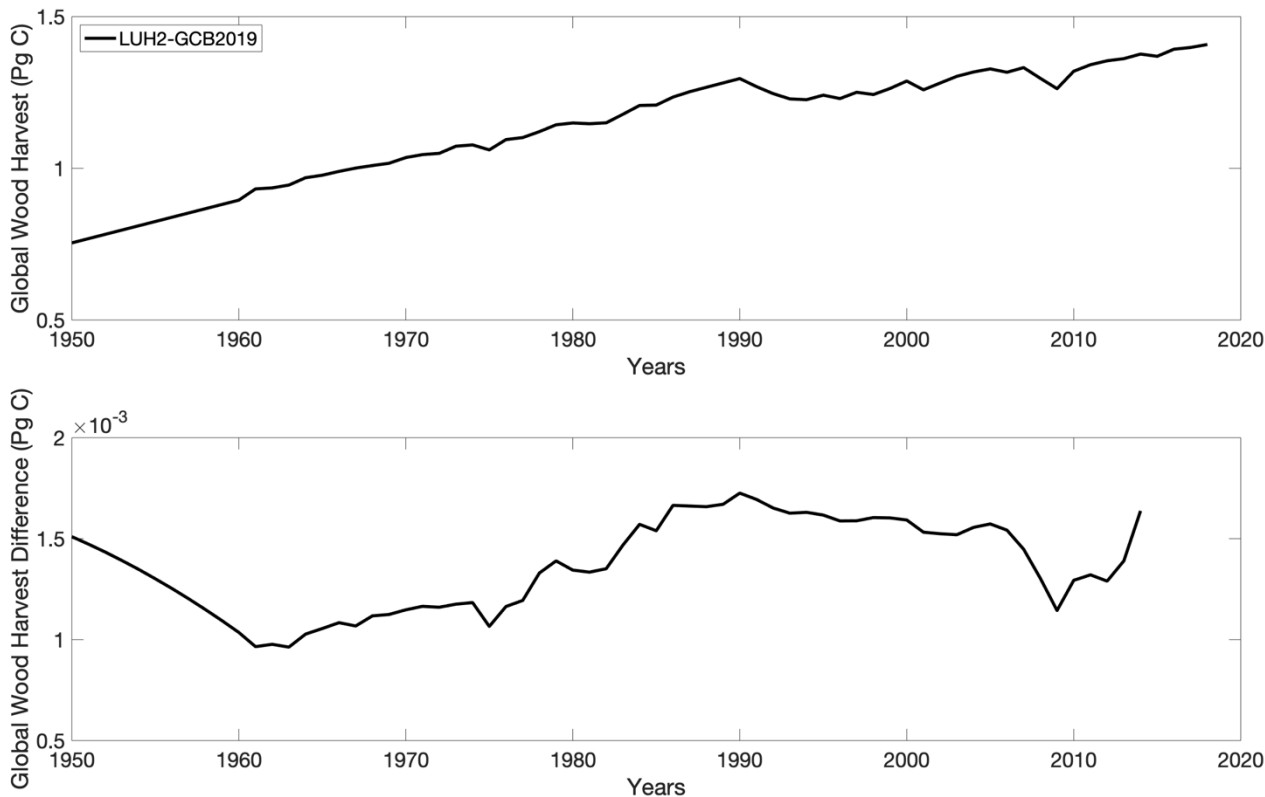


**Figure 3. a) Global wood harvest for LUH2-GCB2019, b) difference between global wood harvest of LUH2 v2h and LUH2-GCB2019 (LUH2 v2h – LUH2-GCB2019)**

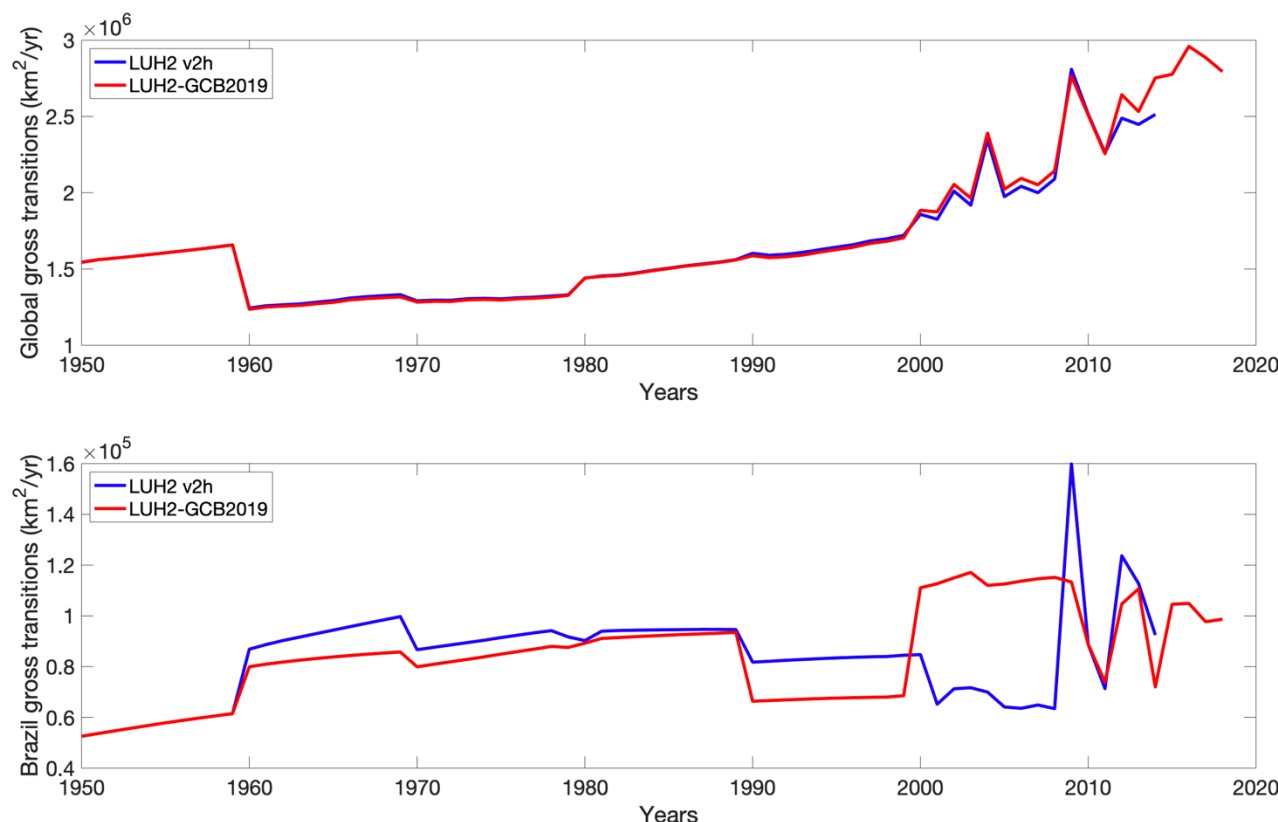

**Figure 4. a) Global and b) Brazil annual gross transitions, for LUH2 v2h and LUH2-GCB2019 for the years 1950-2014/2018**


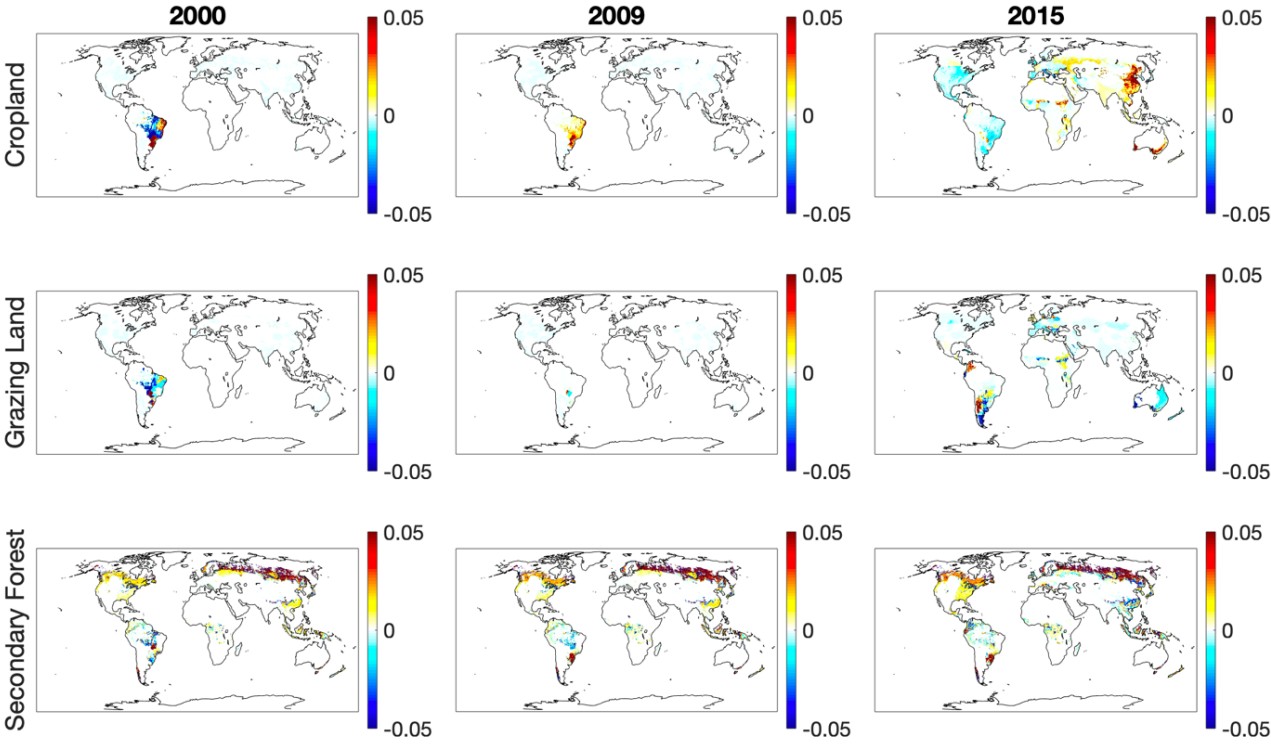

**Figure 5. Mapped differences (LUH2-GCB2019 – LUH2 v2h) of 0.25° grid-cell fractions of cropland, grazing land, and secondary forest in 2000, 2009, and 2015.**