# Peer review of "Land-Use Harmonization Datasets for Annual Global Carbon Budgets"

_Earth System Science Data, 2020_

## Author Response (AR1)

**Reviewer Comments 1:**

General comment:
This paper describes the method for producing the LUH2-GCB datasets, as well as specific details of the recent changes made to the LUH2-GCB2019 dataset. LUH2-GCB2019 dataset corrects cropland and grazing area errors in the underlying input datasets for the globally important region of Brazil, as far back as 1950. This is a very valuable dataset used by both bookkeeping models and Dynamic Global Vegetation Models (DGVMs) for the annual Global Carbon Budget (GCB) assessments.

Response: Thank you for your summary and assessment of our manuscript.

Specific comments:
Sect. 2.3, page 7, line 197-200: For LUH2-GCB2019, a significant update was performed to correct an error in the previously used input data, especially for the country of Brazil. What's the sources of these new data that used for the correction? Does the updated cropland or grazing land data in Brazil based on FAO or HYDE?

Response: The sources for these corrections are listed later in that same paragraph and include updated FAO and HYDE data, as well as ESA satellite data and data from Brazil's IGBE.

Changes: We have added some new wording at the beginning of the paragraph to make these data sources clearer.

Page 5, line 135-137: there are some minor corrections to the method for determining the spatial patterns of wood harvesting. What specific corrections? Or is there any references information?

Response: At the time the LUH2-GCB2019 dataset was produced, some small corrections had been made to the original LUH2 wood harvest algorithm. These corrections resulted in slightly different spatial patterns of primary and secondary wood harvest within each country. These spatial pattern changes were a result of correcting a rounding error in the original algorithm that was used to determine how far from existing human activity the wood harvest patterns were likely to extend. Since these corrections were not a main focus of this paper, we have opted to focus the discussion on the changes in spatial patterns of cropland and grazing land, specifically within Brazil, that were updated for the GCB 2019 studies.

Please define acronym FAO, HYDE on first usage. The authors may need to follow this practice throughout the manuscript.

Response: Thank you for catching these oversights.

Changes: We have added definitions at first usage for the acronyms FAO, HYDE, CMIP6, and checked for other instances as well.

Table 1 needs better presentation, the HYDE simulation years should be given as well.

Response and changes: The table formatting has been updated to reflect design characteristics of tables in other Copernicus publications. An additional column has also been added to the table to provide the HYDE simulation years.

**Reviewer Comments 2:**

Louise Chini et al. "Land-Use Harmonization Datasets for Annual Global Carbon Budgets"
GENERAL:
This paper discussed the methodologies used to produce LUH2-GCB updates and extensions, and described specific details of LUH2-GCB2019 dataset and comparisons with previous LUH datasets. As a newest update, LUH2-GCB2019 dataset made some changes compared with previous datasets, especially correcting cropland and grazing areas of Brazil. This paper is very timely and important for users of this LUH2-GCB2019 or previous versions to better select land use data as input for state-of-the-art models in well-organized scientific assessments. I have only minor comments.

Response: Thank you for your summary and assessment of our manuscript.

My main concern is the capability of LUH2-GCB2019 data capturing regional trends of forest changes. As shown in Figure 5, LUH2-GCB2019 has greater secondary forest areas in mid-high latitudes but lower areas in low latitudes than previous LUH2 v2h dataset over all key time points (2000, 2009 and 2015). These large differences has been related to the changes of wood harvesting used in LUH2-GCB2019, but the changes of spatial patterns on primary forests have not been shown in Figure 5. Please add this information into Figure 5 and further discuss whether this update on secondary forest is better than previous versions especially over China and India, where the recent studies by Chen et al. (2019) showing leading greening tendency. The greenness in China is mainly caused by the afforestation while that in India is likely related to expansion of cropland. Can LUH2-GCB2019 capture those trends?

Response: Thank you for these helpful suggestions.

Changes: We have now included primary forest difference maps in Figure 5, along with a brief discussion of the patterns observed in those maps.

Response: As noted in the manuscript, the LUH2 datasets provide land-use information rather than land cover information, and as a result are not guaranteed to capture regional patterns of land cover changes as observed in remote sensing data. We do include a simple representation of forested vs. non-forested natural vegetation, but this is also defined by the intended use of the land, i.e. a recently harvested forest that is just beginning to regrow would still be defined as a forest in our dataset, even though it does not yet contain any tree cover. However, to the extent that the land cover changes are already well-represented in the FAO data and HYDE data

used by LUH2 as inputs, the LUH2 land-use changes will be consistent with those land cover changes.

Changes: We have added a short paragraph about these features of LUH2 to the manuscript, citing the paper you mentioned as an example.

Minor points:
Some acronyms need to be defined such as FAO, HYDE, BLUE and more at their first appearance.

Response and changes: We have added definitions at first usage for the acronyms FAO, HYDE, BLUE, CMIP6, and checked for other instances as well.

Figure 5: The unit is fraction or %? Please add it into caption.

Response: The units are fractions of gridcell.

Changes: The figure caption has been reworded to hopefully make this clearer to the reader.

Reference
Chen, C. and coauthors: China and India lead in greening of the world through land-use management, Nature Sustainability, 2, 122-129, 2019.

Response and changes: We have added this useful paper to our list of references and cited it in the text.